# WaveFake: A Data Set to Facilitate Audio Deepfake Detection

**Joel Frank**
Ruhr University Bochum
Horst Görtz Institute for IT-Security
joel.frank@rub.de

**Lea Schönherr**
Ruhr University Bochum
Horst Görtz Institute for IT-Security
lea.schoenherr@rub.de

## Abstract

Deep generative modeling has the potential to cause significant harm to society. Recognizing this threat, a magnitude of research into detecting so-called "Deepfakes" has emerged. This research most often focuses on the image domain, while studies exploring generated audio signals have— so far —been neglected. In this paper, we aim to narrow this gap. We present a novel data set, for which we collected ten sample sets from six different network architectures, spanning two languages. We analyze the frequency statistics comprehensively, discovering subtle differences between the architectures, specifically among the higher frequencies. Additionally, to facilitate further development of detection methods, we implemented three different classifiers adopted from the signal processing community to give practitioners a baseline to compare against. In a first evaluation, we already discovered significant trade-offs between the different approaches. Neural network-based approaches performed better on average, but more traditional models proved to be more robust.

## 1 Introduction

$243,000 were lost when criminals used a generated voice recording to impersonate the CEO of a UK company [83]. This is just one of several reports where current state-of-the-art generative modeling was used in harmful ways. Other examples include: impersonation attempts [21], influencing opposition movements [44], being used to justify military actions [26, 55], or online harassment [10]. While there is a multitude of beneficial use cases, for example, enhancing data sets for medical diagnostics [19, 23] or designing DNA to optimize protein bindings [35], finding effective ways to detect fraudulent usage is of utmost importance to society.

Detection in the image domain has received tremendous attention [50, 54, 99, 86, 91, 52, 56, 51, 18, 22]. However, the audio domain is severely lacking. In this paper, we aim at closing this gap. We start by reviewing standard signal processing techniques used for analyzing audio signals. For example, we give an introduction to spectrograms, which are commonly used as an intermediate representation for generative models [43, 67, 96, 97]. Additionally, we provide a survey of current state-of-the-art generative models.

Our main contribution is a novel data set. We collected ten sample sets from six different network architectures across two languages. This paper focuses on analyzing samples that resemble (i.e., recreate) the training distributions. This allows for one-to-one comparisons of audio clips between the different architectures. In this comparison, we find subtle differences between the generators. We also expect good performing classifiers to transfer well to other contexts since recreating the training distribution should yield the most quality samples. We test this hypothesis by also generating completely novel phrases.

35th Conference on Neural Information Processing Systems (NeurIPS 2021) Track on Datasets and Benchmarks.

Finally, we implement three classifiers, which we adopted from best practices in the signal processing community [74, 85], to give future researchers a baseline to compare against [1]. In a first evaluation we already discovered trade-offs between the different approaches. Neural networks performed better overall, but proved to be susceptible to changing settings. Finally, we implemented BlurIG [95] a popular attribution method/package, so practitioners can inspect their predictions when building on our results.

We summarize our main contributions as follows:

- A novel data set consisting of samples from several SOTA network architectures. Additionally, we perform a comprehensive analysis of this data set and find subtle differences between the different architectures.
- An implementation of two baseline models for future researchers to compare against. These models were evaluated in three different settings and we provide a popular attribution method to inspect the prediction.

## 2 Background

In this section, we provide an introduction to standard techniques used for analyzing speech audio signals. For additional material on the topic, the interested reader is referred to the excellent books by Rabiner et al. [70] or Quatieri [69]. Additionally, we provide a survey on current SOTA generative models and summarize related work.

### 2.1 Analyzing speech signals

**(Mel) spectrograms:** A spectrogram is a visual representation of the frequency information of a signal over time (cf. Section 3, Figure 2 for an example). To calculate a spectrogram for an audio signal, we first divide the waveform into *frames* (e.g., 20 ms) with an overlap (e.g., 10 ms) between two adjacent frames. We then apply a window function to avoid spectral leakage [2]. These functions (e.g., Hamming, Hann, Blackman window) are a trade-off between frequency resolution and spectral leakage. Their choice depends on the task and the signal properties, cf. Prabhu [64] for a detailed overview. The frames are then transformed individually using the *Discrete Fourier Transform* (DFT) to obtain a representation in the frequency domain $X(t, k)$. Where $t = 1, \ldots, T$ is the frame index of the signal and $k = 0, \ldots, K-1$ are the DFT coefficients. Finally, we calculate the squared magnitude $|X(t, k)|^2$ of the complex-valued signal to obtain our final representation—the spectrogram.

A commonly used variant is the so-called Mel spectrogram. It is motivated by studies that have shown that humans do not perceive frequencies on a linear scale. In particular, they can detect differences in lower frequencies with a higher resolution when compared to higher frequencies [105]. The Mel scale is an empirically determined non-linear transformation that approximates this relationship:

$$f_{\text{mel}} = 2595 \cdot \log_{10}\left(1 + \frac{f}{700}\right), \tag{1}$$

where $f$ is the frequency in Hz and $f_{\text{mel}}$ the Mel-scaled frequency. To obtain the Mel spectrogram, we apply an ensemble of $S$ triangular filters $H_{\text{mel}}$ (we provide a visual representation in Section 8 of the supplementary material). These filters have a linear distance between the triangle mid frequencies in the Mel scale, which results in a logarithmic increasing distance of the frequencies in the frequency domain

$$X_{\text{mel}}(t, s) = \sum_{k=0}^{K-1} |X(t, k)| H_{\text{mel}}(s, k) \quad \forall s = 1, \ldots, S, \tag{2}$$

which gives us the final Mel spectrogram. Based on it, we can compute a common feature representation for audio analysis.

---

[1] Our code and pretrained models can be found at `github.com/RUB-SysSec/WaveFake`
[2] Energies from one frequency leak into other frequency bins.

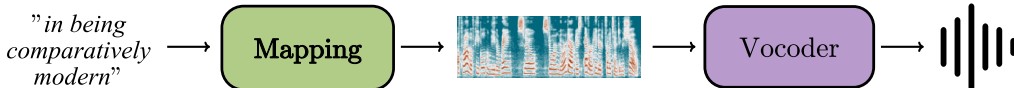

Figure 1: **A typical TTS pipeline.** One model takes a textual prompt with the desired audio transcription (we call it the "mapping" model) and outputs an intermediate representation, for example Mel spectrograms. This intermediate representation is then fed to a second model (the "vocoder") to obtain the final raw audio.

**Mel Frequency Cepstral Coefficients:** *Mel Frequency Cepstral Coefficients* (MFCC) are derived from a Mel-scaled spectrogram by applying a *Discrete Cosine Transform* (DCT) to the logarithm of the Mel-filtered signal

$$c(t, r) = \sum_{s=0}^{S-1} \log \left[ X_{\mathrm{mel}}(t, s) \right] \cdot \cos \left[ \frac{\pi \cdot r \cdot (s + 0.5)}{S} \right] \quad \forall r = 0, \ldots, R - 1, \qquad (3)$$

where $R$ is the number of DCT coefficients.

**Linear Frequency Cepstral Coefficients:** We can also calculate *Linear Frequency Cepstral Coefficients* (LFCC). As the name suggests, these coefficients are derived by applying a linear filterbank (instead of a Mel filterbank) to the signal's spectrogram. This results in retaining more high-frequency information. Except for replacing the filter bank, all other steps remain the same as for MFCC features.

**(Double) Delta features:** MFCCs and LFCCs are often augmented by their first and second derivatives to represent the temporal structure of the input. These are referred to as delta and double delta features, respectively. In practice, these are often calculated by central difference approximation via

$$d(t) = \frac{\sum_{n=1}^{N} n \cdot \left[ c(t + n) - c(t - n) \right]}{2 \cdot \sum_{n=1}^{N} n^2} \quad \forall t = 0, \ldots, T - 1, \qquad (4)$$

where $d(t)$ is the delta at time $t$ and $N$ is a user-defined window length for computing the delta, and $c$ is either the MFCCs/LFCCs or the delta features (when calculating the double delta features).

## 2.2 Text-to-speech (TTS)

In this Section, we want to give a broad overview of different research directions for *Text-To-Speech* (TTS) models. Due to the rapid developments of the field, this is a non-exhaustive list.

While there has been some research into end-to-end models [17, 84], typical TTS models consist of a two-stage approach, represented in Figure 1. First, we enter the text sequence which we want to generate. This sequence gets mapped by some model (or feature extraction method) to a low-dimensional intermediate representation, often linguistic features [8] or Mel spectrograms [58]. Second, we use an additional model (often called vocoder) to map this intermediate representation to raw audio. We focus on the literature on vocoders since it directly connects to our work.

Today, vocoders are typically implemented by Deep Neural Networks (DNNs). The first DNN [101, 20] approaches adopted the parametric vocoders of earlier HMM-based models [102, 88, 98]. Here the DNN was used to predict the statistics of a given time frame, which are then used in traditional speech parameter generation algorithms [88]. Later variants replaced each component in traditional pipelines with neural equivalents [8, 7, 71, 72, 92, 4]. The first architectures which started using DNNs exclusively as the vocoder were auto-regressive generative models, such as WaveNet [58], WaveRNN [33], SampleRNN [53], Char2Wav [82] or Tactron 2 [79].

Due to their auto-regressive nature, these models do not leverage the parallel structure of modern hardware. There have been several attempts to circumvent this problem: One direction is to distill trained auto-regressive decoders into flow-based [38] convolutional student networks, as done by

Parallel WaveNet [58] and Clarinet [62]. Another method is to utilize direct maximum likelihood training as done by several flow-based models, for example, WaveGlow [67], FloWaveNet [36] or WaveFlow [63]. Other probabilistic approaches include those based on variational auto-encoders [59, 61] or diffusion probabilistic models [42, 13]. Another family of methods is based on Generative Adversarial Networks (GANs) [24], examples include, MelGAN [43], GAN-TTS [9], WaveGAN [16], HiFi-GAN [41], Parallel WaveGAN [96] or Multi-Band MelGAN [97].

## 2.3  Related Work

Several previous proposals have collected Deepfake data: The FaceForensics++ dataset [73] curated 1.8 million manipulated images and provides a benchmark for automated facial manipulation detection. Celeb-DF [49] contains high-quality face-swapping Deepfake videos of celebrities with more than 5,000 fake videos. Dolhansky et al. [15] released the Deepfake detection challenge that contains more than 100,000 videos, generated with different methods.

There exists a multitude of research into identifying GAN-generated images: Several approaches use CNNs in the image domain [50, 54, 99, 86, 91], others use statistics in the image domain [52, 56]. Another group of systems employs handcrafted features from the frequency domain: steganalysis-based features [51], spectral centroids [90] or frequency analysis [104, 18, 22, 68]. Li and Lyu [48] proposed a CNN-based Deepfake video detection framework that utilizes artefacts that are consequences of the generation process. Another strain of research combines image analysis with audio analysis. Chintha et al. [14] combined a Deepfake detection with an audio spoofing detection to identify fake videos. At the time of writing and to the best of our knowledge no work has provided an overview over Deepfake audio in isolation.

The signal processing community undertakes a related line of research. The biyearly ASVspoof challenges [94, 87, 57] promote countermeasure against spoofing attacks that aim to fool speaker verification systems via different kinds of attacks. Their benchmarking data sets include replay attacks, voice conversion, and synthesized audio files. Note that the 2021 edition of the challenge features an audio Deepfake track but does not provide specific training data. We imagine our data set to be used complementary with the training data of the challenge. At the time of writing the 2021 edition is still on-going, but evaluating the best performing models in conjunction with our data set is an interesting direction for future work. In the meantime, we adopt one of the baseline models of the ASVspoof challenge to enable a direct comparison. This bi-yearly challenge has led to several proposed models for detecting spoofing attacks, for example, CNN-based methods [89, 46, 45], ensemble methods on different feature representations [60, 34, 76] or methods which detect unusual pauses in human speech [103, 3]. Additionally, another data set is proposed by Kinnunen et al. [39]. They released a re-recorded version of the RedDots database for replay attack detection.

## 3  The data set

In this Section we provide an overview of our data set. It consists of 117,985 generated audio clips (16-bit PCM wav) and can be found on zenodo [3]. In total, it consists of approximately 196 hours of generated audio files. We mostly base our work on the LJSPEECH [31] data set. While TTS models often get trained on private data sets, LJSPEECH is the most common public data set among the publication listed in Section 2.2. Additionally, we consider the JSUT [81] data set, a Japanese speech corpus.

**Reference data:**  We examine multiple networks trained on two reference data sets. First, the LJSPEECH [31] data set consisting of 13,100 short audio clips (on average 6 seconds each; roughly 24 hours total) read by a female speaker. It features passages from 7 non-fiction books, and the audio was recorded with a MacBook Pro microphone. Second, we include samples based on the JSUT [81] data set, specifically, the basic5000 corpus. This corpus consists of 5,000 sentences covering all basic kanji of the Japanese language. (4.8 seconds on average; roughly 6.7 hours total). A female native Japanese speaker performed the recordings in an anechoic room. Note that we do not redistribute the reference data. They are freely available online [31, 81].

---

[3] `https://zenodo.org/record/5642694` - DOI: 10.5281/zenodo.5642694

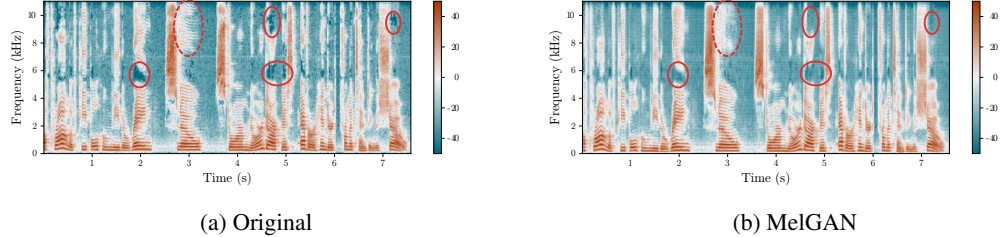

|   (a) Original   |   (b) MelGAN   |

Figure 2: **Spectrograms for the same sample, for different generating models.** They show the frequencies of a signal, plotted over the time of a signal. Lower frequencies at the bottom, higher at the top. Best viewed in color.

**Architectures:** We included a range of architectures in our data set:

- **MelGAN**: We include MelGAN [43], which is one of the first GAN-based generative models for audio data. It uses a fully convolutional feed-forward network as the generator and operates on Mel spectrograms. The discriminator combines three different discriminators that operate on the original and two downsampled versions of the raw audio input. Additionally, it uses an auxiliary loss over the feature space of the three discriminators.

- **Parallel WaveGAN (PWG)**: WaveNet [58] is one of the earliest and most common architectures, We include samples from one of its variants, Parallel WaveGAN [96]. It uses the GAN training paradigm, with a non-autoregressive version of WaveNet as its generator. In a similar vein to MelGAN, it uses an auxiliary loss, but in contrast, matches the *Short-Time Fourier Transform* (STFT) of the original training sample and the generated waveform over multiple resolutions.

- **Multi-band MelGAN (MB-MelGAN)**: Incorporating more fine-grained frequency analysis, might lead to more convincing samples. We include MB-MelGAN, which computes its auxiliary (frequency-based; inspired by PWG) loss in different sub-bands. Its generator is based on a bigger version of the MelGAN generator. Still, instead of predicting the entire audio directly, the generator produces multiple sub-bands, which are then summed up to the complete audio signal.

- **Full-band MelGAN (FB-MelGAN)**: We include a variant of MB-MelGAN which generates the complete audio directly and computes its auxiliary loss (the same as PWG) over the full audio instead of its sub-bands.

- **HiFi-GAN (HiFi-GAN)**: HiFi-GAN [41] utilizes multiple sub-discriminators, each of which examines only a specific periodic part of the input waveform. Similarly, its generator is built with multiple residual blocks, each observing patterns of different lengths in parallel. Additionally, HiFi-GAN employs the feature-space-based loss from MelGAN and minimizes the $L_1$ distance between the Mel spectrogram of a generated waveform and a ground truth one in its loss function.

- **WaveGlow**: The training procedure might also influence the detectability of fake samples. Therefore, we include samples from WaveGlow to investigate maximum-likelihood-based methods. It is a flow-based generative model based on Glow [37], whose architecture is heavily inspired by WaveNet.

Additionally, we examine MelGAN both in a version similar to the original publication, which we denote as MelGAN, and in a larger version with a bigger receptive field, MelGAN (L)arge. This version is similar to the one used by FB-MelGAN, allowing for a one-to-one comparison. Finally, we also obtain samples from a complete TTS-pipeline. We use a conformer [25] to map novel phrases (i.e., not part of LJSPEECH) to Mel spectrograms. Then we use a fine-tuned PWG model (trained on LJSPEECH) to obtain the final audio. We call this data set TTS. In total, we sample ten different data sets, seven based on LJSPEECH (MelGAN, MelGAN (L), FB-MelGAN, HiFi-GAN, WaveGlow, PWG, TTS) and two based on JSUT (MB-MelGAN, PWG).

**Sampling procedure:** For WaveGlow we utilize the official implementation [66] (commit 8afb643) in conjunction with the official pre-trained network on PyTorch Hub [65]. HiFi-GAN also offers a public repository with pretrained models [40]. We use a popular implementation available on GitHub [27] (commit 12c677e) for the remaining networks. When sampling the data set, we first extract Mel spectrograms from the original audio files, using the pre-processing scripts of the corresponding repositories. We then feed these Mel spectrograms to the respective models to obtain

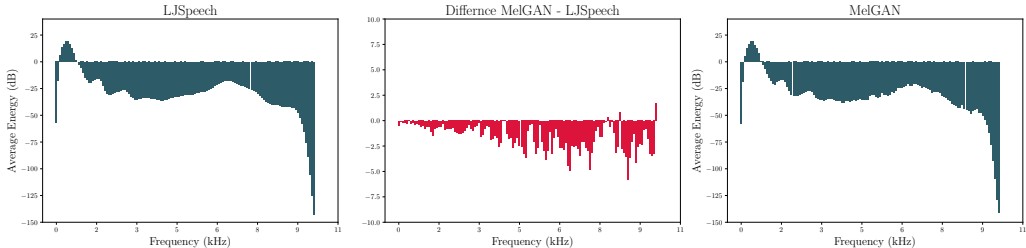

Figure 3: **Average energy per frequency bin.** We show the average energy per frequency bin in dB. Additionally, we plot the difference to the original data (LJSPEECH).

Table 1: **Basic statistics for all LJSPEECH-based data sets.** We report the average pitch frequency and its standard deviation as well as the average spectral centroid.

|  | LJSPEECH | MelGAN | MelGAN (L) | FB-MelGAN | MB-MelGAN | HiFi-GAN | WaveGlow | PWG |
|---|---|---|---|---|---|---|---|---|
| Avg. Pitch | 137.61 | 133.51 | 130.94 | 135.22 | 133.519 | 133.60 | 135.80 | 131.018 |
| Std. Pitch | 49.64 | 47.00 | 46.15 | 48.90 | 48.19 | 48.09 | 47.38 | 47.08 |
| Avg. Centroid | 2367.79 | 2414.81 | 2355.59 | 2362.22 | 2414.81 | 2374.51 | 2422.63 | 2348.31 |

the data set. Intuitively, the networks are asked to "recreate" the original data sets. For sampling the full TTS results, we use the ESPnet project [93, 28, 30, 47]. To make sure the generated phrases do not overlap with the training set, we downloaded the common voices data set [6] and extracted 16,283 phrases from it.

**Differences between the architectures:** We perform an analysis of the differences between the architectures. First, by plotting the spectrograms of an audio file in Figure 2 (LJSPEECH 008-0217; all data sets can be found in Section 6 of the supplementary material). Generally, all architectures produce spectrograms different from the original. The networks seem to struggle with the absence of information generally (solid circles in Figure 2a) as well as with higher frequencies, especially for vocals (dashed circle). Additionally, MelGAN and WaveGlow seem to cause a repeating horizontal pattern. The remaining networks (all using an auxiliary loss in the frequency domain) do not seem to exhibit this behavior. However, they still produce apparent differences. Note that these differences are visible when plotting the audio but generally inaudible when listening to the samples.

Second, we perform a prosody analysis of each data set. We collect 10.000 samples from LJSPEECH and the corresponding sample from each of our architectures. For each data set, we compute the fundamental frequency (pitch) by using normalized cross-correlation and median smoothing [29] in the range $50 - 500$Hz. Additionally, we compute the center of mass of the frequency spectrum by the mean of the frequencies weighted by their magnitudes (the so-called spectral centroid). The results can be found in Table 1 and confirm our visual observations. While all architectures come close to the original, none can approximate it perfectly. Generally, all vocoders produce a lower and less varied pitch.

The spectral centroid results are varied. To investigate further, we perform an additional, more fine-grained analysis by plotting a histogram of the energy contained in each frequency bin. Furthermore, we plot the relative difference to the original data, i.e., the difference weighted by the initial (LJSPEECH) energy. The plots can be found in Figure 3. For brevity, we only show the MelGAN comparison here, the other histograms can be found in Section 6 of the supplementary material. However, all analyses had similar results. The histograms' overall shape is identical, but the generated samples exhibit apparent differences, especially in the higher frequencies.

**Licensing:** The LJSPEECHdata set is in the public domain. The JSUTcorpus is licensed by CC-BY-SA 4.0, noting that redistribution is only permitted in some instances. We contacted the author, who saw no conflict in distributing our fake samples, as long as it's for research purposes. Thus, we also released our data set under a CC-BY-SA 4.0 license.

**Ethical considerations:** Our data set consists of phrases from non-fiction books (LJSPEECH) and everyday conversational Japanese (JSUT), which are already available online. The same is true for all models used to generate this data set. One might wonder if releasing research into detecting

Deepfakes might negatively affect the detection "arms race". This discussion has a long-standing history in the security community, and the general conclusion is that withholding research is hurtful. We provide a more in-depth discussion of this topic from the perspective of security researchers in the supplementary material (Section 1).

## 4   Providing a baseline

We base our experiment on the ASVspoofchallenge [87] introduced in Section 2.3. The challenge aims to promote research into detecting (audio) spoofing attacks and speaker verification. As a point of reference, the challenge offers two baseline models (*Gaussian Mixture Model* (GMM) and RawNet2 [32, 85]). We adopt these models and the metric used by the challenge to compare the two domains.

### 4.1   Experiments

We start by training seven different classifiers, one for each vocoder in our data set (MelGAN, MelGAN (L), FB-MelGAN, MB-MelGAN, HiFi-GAN, PWG and WaveGlow). For training our classifiers, we exclusively use the data sets based on LJSPEECH. We use the JSUT (different speaker, language, and recording setup) and TTS (same speaker, completely novel phrases) data sets for accessing the classifiers ability to generalize to an unknown setting. We train six additional models in a leave-one-out setting to control if the models picked up on vocoder-specific characteristics. Finally, we simulate a phone recording to emulate a real-world fraud attempt.

For each classifier, we evaluate the performance on all vocoders over a hold-out set of 20% of the data. We use the *Equal Error Rate* (EER) as our evaluation metric. The ASVspoof challenge also uses this metric. It is defined as the point on the ROC curve, where false acceptance rate and false rejection rate are equal and is commonly used to assess the performance of binary classifications tasks like biometric security systems [75]. The best possible value is 0.0 (no wrong predictions), worst 1.0 (everything wrong), guessing is 0.5. Additionally, we compute *average Equal Error Rate* (aEER) over all test sets.

When training GMM models, we follow Sahidullah et al. [74] and train two GMMs per data set, one fitting the real distribution (the original LJSPEECH data set) and one fitting the generated audio samples (the respective vocoder-samples from our data set). In addition to the LFCC features used by Sahidullah et al. [74], we evaluate MFCC features. To classify a given sample, we calculate its likelihood $\Lambda(\mathbf{X})$ via

$$\Lambda(\mathbf{X}) = \log p(\mathbf{X}|\theta_n) - \log p(\mathbf{X}|\theta_s), \tag{5}$$

where $\mathbf{X}$ are the input features (namely MFCC or LFCC) and $\theta_n$ and $\theta_s$ are the GMM model parameter of the real and the generated audio distributions, respectively. The out-of-distribution models are exclusively trained on LFCC features, since we found them to strictly outperform the MFCC features (cf., Section 2).

Additionally, we train RawNet2 [32] instances to investigate a neural alternative. RawNet2 is a CNN-GRU hybrid model which extracts a speaker embedding directly from raw audio. When used to perform speaker verification (or Deepfake detection), a fully connected layer is trained on top of this embedding to make the final decision [85]. Details on all training setups can be found in the supplementary material (Section 3).

**Single training set:**   In a first experiment, we evaluate the performance when training on a single data set. For the GMM experiments, we only present the LFCC results since we found them to outperform the MFCC features strictly. LFCC features contain a significantly higher amount of high-frequency components. We hypothesize that these are meaningful for achieving good overall performance. Similar patterns were observed in the image domain [22], implying that methods might transfer between the two. The results are presented in Table 2. The rows show the respective training sets and the columns the different test sets. Gray values indicate that the same generative model is used for the training of the GMM classifier as for the test set.

When training on a single data set, we observe that FB-MelGAN serves as the best prior for all other data sets, achieving the lowest average EER (0.062). Intuitively this makes sense. FB-MelGAN uses the same architecture as MelGAN (L), while deploying a similar auxiliary loss as PWG or

Table 2: **Equal Error Rate (EER) of the baseline classifier on different subset (LFCC).** We train a new GMM model for each data set and compute the EER as well as the **aEER**.

| Training Set | LJSPEECH | | | | | | | | JSUT | | aEER |
|---|---|---|---|---|---|---|---|---|---|---|---|
| | MelGAN | MelGAN (L) | FB-MelGAN | MB-MelGAN | HiFi-GAN | WaveGlow | PWG | TTS | MB-MelGAN | PWG | |
| MelGAN | 0.148 | **0.094** | 0.155 | 0.153 | 0.168 | 0.189 | 0.109 | 0.023 | 0.384 | 0.533 | 0.215 |
| MelGAN (L) | **0.119** | 0.044 | 0.176 | 0.132 | 0.150 | 0.245 | 0.115 | **0.012** | 0.406 | 0.607 | 0.222 |
| MB-MelGAN | 0.359 | 0.316 | 0.002 | 0.124 | 0.083 | 0.007 | **0.017** | 0.021 | 0.017 | 0.051 | 0.108 |
| FB-MelGAN | 0.197 | 0.133 | 0.030 | 0.025 | **0.034** | 0.037 | 0.019 | 0.025 | 0.026 | 0.058 | **0.062** |
| HiFi-GAN | 0.255 | 0.193 | 0.034 | **0.050** | 0.029 | 0.035 | 0.020 | 0.018 | 0.057 | 0.123 | 0.089 |
| PWG | 0.402 | 0.374 | 0.008 | 0.161 | 0.100 | 0.001 | **0.017** | 0.042 | **0.014** | **0.042** | 0.124 |
| WaveGlow | 0.287 | 0.237 | **0.015** | 0.066 | 0.041 | **0.008** | 0.003 | 0.015 | 0.031 | 0.075 | 0.085 |

When the distribution is part of the training set we highlight it in gray. For other results, we highlight the best results per column in **bold**.

Table 3: **Equal Error Rate (EER) of the RawNet2 classifier.** We train a single RawNet2 model per data set and compute the EER as well as the **aEER**.

| Training Set | LJSPEECH | | | | | | | | JSUT | | aEER |
|---|---|---|---|---|---|---|---|---|---|---|---|
| | MelGAN | MelGAN (L) | FB-MelGAN | MB-MelGAN | HiFi-GAN | WaveGlow | PWG | TTS | MB-MelGAN | PWG | |
| MelGAN | 0.001 | **0.001** | 0.485 | 0.509 | 0.525 | 0.497 | 0.407 | 0.356 | 0.113 | 0.089 | 0.292 |
| MelGAN (L) | **0.008** | 0.000 | 0.511 | 0.490 | 0.486 | 0.369 | 0.446 | 0.265 | **0.009** | **0.003** | **0.258** |
| MB-MelGAN | 0.118 | 0.371 | 0.003 | 0.282 | **0.216** | 0.302 | **0.002** | 0.522 | 0.922 | 0.997 | 0.357 |
| FB-MelGAN | 0.161 | 0.239 | 0.122 | 0.082 | 0.304 | **0.259** | 0.130 | 0.391 | 0.974 | 0.994 | 0.363 |
| HiFi-GAN | 0.174 | 0.437 | 0.242 | 0.364 | 0.023 | 0.359 | 0.057 | 0.098 | 0.499 | 0.719 | 0.319 |
| PWG | 0.052 | 0.358 | 0.261 | **0.234** | 0.324 | 0.000 | 0.006 | **0.001** | 0.984 | 0.999 | 0.358 |
| WaveGlow | 0.086 | 0.379 | **0.079** | 0.361 | 0.226 | 0.316 | 0.001 | 0.250 | 0.409 | 0.786 | 0.294 |

When the distribution is part of the training set we highlight it in gray. For other results, we highlight the best results per column in **bold**.

MB-MelGAN. Generally, we can see a clear divide between MelGAN/MelGAN (L) and the other networks, which we will explore in Section 4.2.

When examining completely novel data (JSUT), all classifier drop in performance. However, PWG, WaveGlow, HiFi-GAN, FB-MelGAN and, MB-MelGAN still serve as a good prior, implying that the generating architectures exhibit common patterns which can be recognized for different training data sets and speakers. Again, a similar pattern was also observed in the image domain [91]. The TTS data set is one of the easiest data sets. This undermines our belief that data that recreates the training set is harder to classify correctly. Interestingly the PWG classifier does not generalize well to the TTS data set. Remember that while we use completely novel phrases, the vocoder for this data set is a PWG network trained on LJSPEECH. This might imply that our models overfit their specific training set.

This trend can also be seen in the RawNet2 results, which overall perform worse than the GMM models. They seem to overfit their respective training architecture, which prevents them from generalizing to other data sets. This explains the good performance of the MelGAN/MelGAN (L) models and the PWG/TTS models, since these pairs share generator architectures. Additionally, we can note that the MelGAN/MelGAN (L) models serve as a good prior for generalizing to the JSUT data sets.

**Leave-one-out:** We explore this hypothesis by running a leave-one-out experiment. Results can be found in Table 4. Overall the results improve on the aEER ($0.062 \rightarrow 0.058$). Also, the generalization results to a novel setting (JSUT) increase significantly. However, FB-MelGAN seems to be an essential ingredient for good performance on the JSUT data since not training on it hurts performance significantly. Additionally, the MelGAN and MelGAN (L) data sets still prove to be a challenge, even when included in the training set.

The results are similar for RawNet2 (Table 5). When trained on multiple distributions, the networks can successfully generalize, even surpassing the best aEER (0.04). However, some models still overfit to the training data, making generalization to JSUT impossible (MelGAN (L), MB-MelGAN, WaveGlow). Additionally, the better average performance is traded off with worse performance on the training data. For example, the best performing average model has a 13% false acceptance/false rejection rate. This would be unacceptable in a real-life setting.

**Simulating a phone call:** Finally, we return to our motivating example and examine how well our models generalize to a (simulated) real-life setting. We emulate a phone recording for the three test data set (both JSUT data sets and the full TTS-pipeline) and evaluate the out-of-distribution models on them. The GMM classifiers work exceptionally well, even surpassing the performance in the

Table 4: **Equal Error Rate (EER) for the GMM classifier in an out-of-distribution setting.** We train a new GMM model for each but one distribution on LFCC features.

| | LJSPEECH | | | | | | | | JSUT | | |
|---|---|---|---|---|---|---|---|---|---|---|---|
| Training Set | MelGAN | MelGAN (L) | FB-MelGAN | MB-MelGAN | HiFi-GAN | WaveGlow | PWG | TTS | MB-MelGAN | PWG | aEER |
| MelGAN | 0.220 | 0.146 | 0.009 | 0.051 | 0.040 | 0.016 | 0.009 | 0.006 | 0.023 | 0.067 | 0.065 |
| MelGAN (L) | 0.231 | 0.164 | 0.010 | 0.045 | 0.040 | 0.014 | 0.012 | 0.009 | 0.013 | **0.043** | 0.064 |
| MB-MelGAN | 0.187 | 0.117 | 0.013 | 0.043 | 0.039 | 0.018 | 0.010 | **0.002** | 0.058 | 0.141 | 0.069 |
| FB-MelGAN | 0.191 | 0.116 | 0.013 | 0.058 | 0.053 | 0.022 | 0.013 | 0.003 | 0.084 | 0.220 | 0.086 |
| HiFi-GAN | 0.192 | 0.119 | 0.011 | 0.050 | 0.047 | 0.015 | 0.012 | 0.004 | **0.020** | 0.061 | **0.058** |
| PWG | 0.176 | 0.105 | 0.014 | 0.044 | 0.042 | 0.034 | 0.013 | 0.005 | 0.033 | 0.101 | 0.062 |
| WaveGlow | 0.191 | 0.114 | 0.013 | 0.049 | 0.045 | 0.021 | 0.015 | 0.008 | 0.031 | 0.078 | 0.062 |

When the distribution is part of the training set we highlight it in gray. For other results, we highlight the best results per column in **bold**.

Table 5: **Equal Error Rate (EER) of the RawNet2 classifier in an out-of-distribution setting.** We train a single RawNet2 model on all but one distribution and compute the EER as well as the **aEER**.

| | LJSPEECH | | | | | | | | JSUT | | |
|---|---|---|---|---|---|---|---|---|---|---|---|
| Training Set | MelGAN | MelGAN (L) | FB-MelGAN | MB-MelGAN | HiFi-GAN | WaveGlow | PWG | TTS | MB-MelGAN | PWG | aEER |
| MelGAN | 0.008 | 0.005 | 0.023 | 0.137 | 0.098 | 0.076 | 0.011 | 0.019 | **0.000** | **0.000** | **0.040** |
| MelGAN (L) | 0.005 | 0.046 | 0.009 | 0.048 | 0.050 | 0.024 | 0.004 | 0.020 | 0.985 | 0.996 | 0.241 |
| MB-MelGAN | 0.013 | 0.039 | 0.037 | 0.102 | 0.060 | 0.055 | 0.005 | 0.089 | 0.860 | 0.758 | 0.214 |
| FB-MelGAN | 0.013 | 0.023 | 0.032 | 0.216 | 0.058 | 0.054 | 0.011 | 0.026 | 0.092 | 0.088 | 0.065 |
| HiFi-GAN | 0.006 | 0.009 | 0.031 | 0.113 | 0.196 | 0.065 | 0.010 | 0.044 | 0.001 | 0.001 | 0.048 |
| PWG | 0.005 | 0.004 | 0.026 | 0.108 | 0.088 | 0.209 | 0.011 | 0.044 | 0.047 | 0.123 | 0.069 |
| WaveGlow | 0.001 | 0.006 | 0.008 | 0.038 | 0.046 | 0.010 | 0.001 | **0.005** | 0.828 | 0.904 | 0.205 |

When the distribution is part of the training set we highlight it in gray. For other results, we highlight the best results per column in **bold**.

out-of-distribution setting. The highest EER we could detect was 0.003, all other results were lower or separated the data perfectly (a full table can be found in Section 5 of the supplementary material). The results are flipped for the RawNet2 models. While in the clean setting, the (some) models classify the data almost perfectly, under the phone simulation, the error rate shoots up significantly.

While we only examined a simulated setting, we take this as the first evidence that our data set can be used to extrapolate classifier performance to the real world. Overall, we can conclude that these first results are encouraging, but there is still much room for improvement.

### 4.2 Attribution

Finally, we want to investigate which parts of the audio signal influence the prediction. To this end, we implemented BlurIG [95], a popular attribution method. We inspect the attribution of four classifiers for the audio clip used in Section 3. The results are displayed in Figure 4.

Overall, we can see a shift from very broad attention, spread somewhat evenly across all three feature representations (MelGAN (L)), to more narrow-focused attention across very specific filters (PWG). MelGAN (L) and FB-MelGAN classifiers operate (mostly) on the higher frequencies, while MB-MelGAN and PWG also rely on low frequencies for the detection. These observations confirm our suspicion about the MFCC features. They mask higher frequencies, needed for classifying MelGAN (L) and FB-MelGAN, while over-representing lower frequencies, which still leads to a good performance on the MB-MelGAN and PWG data sets. This also explains the performance of FB-MelGAN, which strikes a balance between all necessary features. The spread-out attribution might also explain the poor in-distribution performance of the classifiers trained on the MelGAN variants since the classifier needs to focus on a broader range of features.

## 5 Discussion

In this paper, we took the first step towards research into audio Deepfakes. While we hope our data set proves useful for future practitioners, there are several limitations to our work:

**Evaluating on realistic data:** The difficulties of obtaining realistic data set have been a long-standing problem in the security community [80]. Often benign data is readily available, but data used in malicious contexts is hard to come by. That leaves us with estimating real-world performance on proxy data. We argue that in our case, we might have good odds that results transfer. Currently,

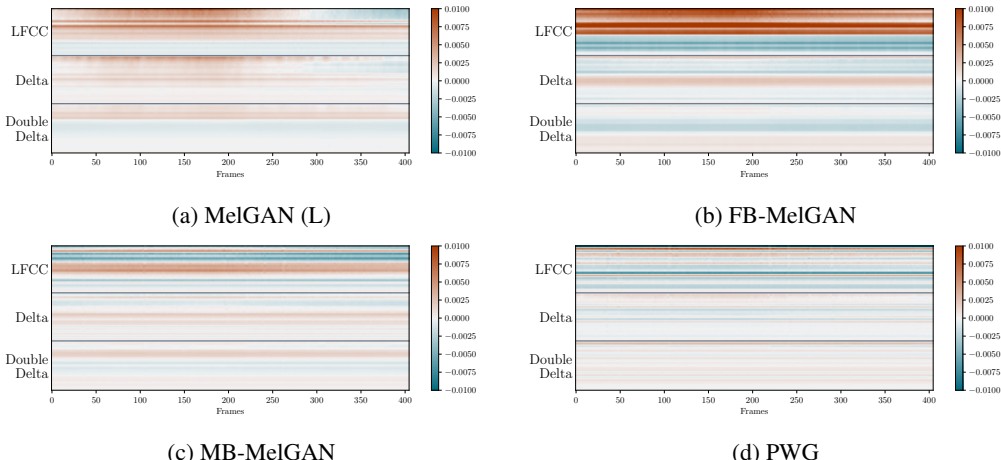

|  | (a) MelGAN (L) | | (b) FB-MelGAN |
|---|---|---|---|
|  | (c) MB-MelGAN | | (d) PWG |

Figure 4: **Attribution of the different models on a real audio sample.** We show the LFCC, delta, and, double delta features. The plot can be read similarly to the spectrogram plots, i.e., features computed over lower frequencies are at the bottom of their respective section, features over higher frequencies are at the top. Best viewed in color.

images generated by off-the-shelf neural networks are used in malicious attempts [10]. We expect the number of audio Deepfakes to increase as well.

**Variety of the data:** We specifically choose to focus on the LJSPEECH corpus since it is commonly used for training generative audio models. That allows a one-to-one comparison. However, it only contains recordings by one speaker. We can make some observations about generalization by comparing against the JSUT and TTS data sets, but a broader analysis focusing on different scenarios would be ideal.

**Adversarial examples and perturbations:** Deepfake-image detectors have already been shown to be vulnerable against adversarial examples [11]. There also exists a myriad of adversarial attacks against automatic speech recognition [12, 77, 100, 77, 5, 78, 2] (Abdullah et al. [1] provide a survey). We have looked at phone recordings, but classifiers should report their robustness against these attacks and other common perturbations (noise, room responses, over-the-air settings, etc.) as part of their evaluation. In this work, we focused on providing the first steps towards audio Deepfake detection.

## 6 Conclusion

This paper presents a starting point for researchers who want to investigate generated audio signals. We started by presenting a broad overview of signal processing techniques and common feature representations as well as a survey of the current TTS landscape. We then moved on to introduce our main contribution, a novel data set, with samples from six different state-of-the-art architectures across two languages. We discovered subtle differences between the different models by plotting the frequency spectrum, especially among the higher frequencies. Following up, we conducted a prosody analysis and investigated each data set's average energy per frequency. This analysis confirmed our previous findings, revealing that while all models come close to correctly approximating the training data, we can still detect differences unique to each model. To provide a baseline for future practitioners, we trained several baseline models. We evaluated their performance across the different data sets and multiple settings. Specifically, we trained GMM and neural network-based solutions. While we found the neural networks to perform better overall, the GMM classifiers proved to be more robust, which might give them an advantage in real-life settings. Finally, we inspected the different classifiers using an attribution method. We found that lower frequencies cannot be neglected while high-frequency information proved indispensable.

## Acknowledgments and Disclosure of Funding

We would like to thank our colleagues Thorsten Eisenhofer, Thorsten Holz, Dorothea Kolossa, and our anonymous reviewers for their valuable feedback and fruitful discussions. Additionally, we would like to thank Tomoki Hayashi, Hemlata Tak, and the WaveGlow team for their excellent repositories. This work was supported by the Deutsche Forschungsgemeinschaft (DFG, German Research Foundation) under Germany's Excellence Strategy – EXC-2092 CASA – 390781972.

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
