# OpenReview forum: "WaveFake: A Data Set to Facilitate Audio Deepfake Detection"
_NeurIPS.cc/2021/Track/Datasets_and_Benchmarks/Round2 — NeurIPS 2021 Datasets and Benchmarks Track (Round 2)_

### Official Review · Reviewer_L9oa · 2021-09-03
**A significant and commendable effort, with a few areas of improvement to strengthen its overall impact**

**Rating:** 7
**Confidence:** 4

**Strengths:**

The main strength of this submission is its completeness and detail. It is clear by looking at the code, the appendix, and the dataset repository on Zenodo, that a lot of effort has been put into producing this work. Therefore, the dataset is easily accessible and will continue to be. I believe the contribution is also important, as, like the authors state, audio applications tend to be neglected with overwhelming focus placed on vision methods. In particular, this dataset is used to help facilitate model development for spoof detection tasks such as ASVspoof, which provides an evaluation dataset but no data for training.

**Weaknesses:**

The main weakness is the quality of writing, with a considerable number of grammatical mistakes present which affect readability. It should be noted however that the authors are not native English speakers. Furthermore, the abstract and conclusion undermine the quality of work of the submission. Improvement in those sections in particular will add polish to the paper and strengthen its impact  (see further detail in Clarity for all these points).

I will add a few points here:
1. I personally find it difficult to understand how exactly the baseline challenges relate to the ASVspoof challenge - I was able to get an understanding from the reference [83] on line 231, but a short paragraph explaining the links/the challenge itself clearly in lines 230+ would be very helpful to new readers.

2. Section 2.1 is listed as a key contribution
> First, we provide researchers with an introduction to common signal processing techniques used for analyzing audio signals

    but this information is available elsewhere with better clarity. In my opinion the limited space is better used for a more thorough review of relevant methods and more detailed explanation of TTS systems/audio deepfake methods (or even bringing some of the results from the appendix into the main text e.g. phone call simulation). Alternatively, the authors could expand on their
    > survey of current state-of-the-art generative models.

    and still keep their contributions as a list of 3 (survey, data, baselines).

**Additional Feedback:**

I believe the previous reviewers’ criticisms were appropriately tackled, and I would be willing to increase my score if the authors show evidence of being receptive to implementing some further issues and suggestions raised in this review.

Please be aware than the link to the past reviews does not give me access to view them, so if I contradict any individual comments made there, it was because I have no way of seeing the full reviews and rebuttals. Perhaps the area chairs can open those up if we need to discuss any points from there further.

# Update: Post rebuttal

I have raised my score to 7 and increased confidence to 4 based on the author's responses during the rebuttal period. A few outstanding changes remain, which I would like to see implemented before the camera ready deadline to keep this score. Many thanks to the authors!

**Clarity:**

This paper is in need of improvement in the clarity of writing. I will offer a list of suggestions to the authors which may change my score during the rebuttal phase:

1. Please correct the following mistakes:
    * Line 22: Additionally, **we** provide a survey of current state-of-the-art 24 generative models.
    * Line 34: .. BlurIG, a popular attribution methods **module**/**package**.
    * Line 43: refereed -> **referred** (also appears in other places in text)
    * Line 99: The first architectures who -> architectures **which**
    * Line 197: generally struggle with the absent->**absence** of information
    * Line 226: This is a long standing debate in the security community and the overall consensus and we have provided a thorough discussion in the supplementary material. -> ?
    * Line 273: This undermines our believe -> **belief** that data
    * Line 314: As of right know -> **now**

2. The abstract and conclusion can be strengthened to really reflect the quantity and quality of work put into this submission (see suggestions):
    * Abstract:
> Finally, we supply practitioners with two baseline models, adopted from the signal processing community, to facilitate further research in this area.

      Can you be more specific as to what these baseline models exactly achieve/motivate?

  * Conclusion:  In each sentence, once again, can you more specifically describe the results you were able to achieve? (e.g. *"In a first analysis, we already discovered subtle differences between the different models, especially among the higher frequencies"*: analysing with what, what were the differences etc.?)

3. Upon each reference to material in the Appendix, can you please link specific figures and sections instead of writing *“can be found in the supplementary material”*?

**Correctness:**

The experiments to the best of my knowledge are built in a sound way: the baseline tasks were based on metrics used by the ASVspoof challenge and built to assist in training models for the task specifically. The main body of text and code supplied accurately reflect the claims made by the authors in terms of contributions.

**Documentation:**

The data is easily accessible on Zenodo and sufficiently documented, thus ensuring reliable access over time. Code was supplied to reviewers and appears to be of high quality. I would like to see the code more clearly linked also in the datasheet (I believe the GitHub link is currently inaccessible, please make this available when ready).

One criticism is that the datasheet is not very detailed, and I would recommend the authors complete a datasheet as suggested in the call for papers (e.g. [Datasheet for Datasets](https://arxiv.org/abs/1803.09010) as the most easily accessible version).

**Ethics:**

A further discussion on ethics was prompted by the previous reviewers. I believe the authors have demonstrated sufficient changes from the previous version to alleviate any ethical concerns, and I agree with their statement that “security by obscurity” is not a good strategy for defence against attacks. As such I have no issue to the best of my knowledge with the ethical implications of this work. All source material is appropriately credited and permission where necessary was obtained.

**Relation To Prior Work:**

The related work section identifies a scope of contribution for this work, primarily as training data for various spoof detection challenges (e.g. ASVspoof). As this is not exactly my field of expertise, I will defer to comments of other reviewers on whether there are any particular works missing. Beyond this, this section seems comprehensive to me.

**Summary And Contributions:**

This paper is a resubmission from Round 1. The paper introduces datasets for the purpose of audio deepfake detection. The dataset is approximately 175 hours of generated audio files, based on the LJSPEECH and JSUT datasets. Nine sample sets from five different network architectures were collected. The authors also provide an adopted baseline from the bi-yearly ASVspoof challenge.

This paper contributes a step towards facilitating research in audio deepfake detection, which is an emerging but important field. Experiments conducted were comprehensive and should be of interest to the deepfake/audio deep learning communities.

---

> ### Author Response · Authors · 2021-09-30
> **Reviewer L9oa**
>
> We thank the reviewer for this in-depth feedback. We have taken several steps outlined below to address the issues pointed out in the review.
>
> > I personally find it difficult to understand how exactly the baseline challenges relate to the ASVspoof challenge
>
> We have added a corresponding explanation.
>
> > Section 2.1 is listed as a key contribution
>
> As outlined above, we have adjusted the corresponding claim and used the space to strengthen other sections.
>
> > Please correct the following mistakes:
>
> We have corrected the listed mistakes and generally proofread and corrected the paper.
>
> > The abstract and conclusion can be strengthened to really reflect the quantity and quality of work put into this submission
>
> We have rewritten the abstract and the conclusion accordingly.
>
> > Upon each reference to material in the appendix, can you please link specific figures and sections
>
> We added a reference in the main paper.
>
> > One criticism is that the datasheet is not very detailed, and I would recommend the authors complete a datasheet as suggested in the call for papers (e.g. [Datasheet for Datasets](https://arxiv.org/abs/1803.09010) as the most easily accessible version).
>
> Please note that we already used the framework in our submission. We simply choose to answer the questions in full text instead of the question-based format. We agree that consistency is key here and have revamped our datasheet.
>
> > Please be aware that the link to the past reviews does not give me access to view them, so if I contradict any individual comments made there, it was because I have no way of seeing the full reviews and rebuttals. Perhaps the area chairs can open those up if we need to discuss any points from there further.
>
> We assumed reviewers would be given access to the previous round of reviews. We contacted the PC chairs, and they pointed out that this is intentional since the second round of reviews should happen independently.

---

> > ### Comment · Reviewer_L9oa · 2021-09-30
> > **Re: Changes**
> >
> > Thanks for incorporating reviewer feedback:
> >
> > I would tweak the conclusion further:
> >
> > > By plotting the frequency spectrum, we performed a first visual analysis. In which we discovered subtle differences between the different models, especially among the higher frequencies. Following up, we conducted a prosody analysis of each data set. This analysis confirmed our previous findings, revealing that there are apparent differences while all models come close.
> >
> > to
> >
> > > By plotting the frequency spectrum, we discovered subtle differences between the different models, especially among the higher frequencies.
> >
> > and also elaborate on:
> >
> > > Following up, we conducted a prosody analysis of each data set. This analysis confirmed our previous findings, revealing that **there are apparent differences while all models come close.**
> >
> > The statement in bold still is vague -- imagine reading the conclusion as a standalone section assuming no knowledge of the paper content (It should be fairly self contained and informative that if someone wishes to skim the conclusion, they will be inspired to learn more and also get an idea of the key points to take away from the paper).
> >
> > I am happy with all remaining changes -- regarding the overview section, sometimes you just can't trust reviewers! I wouldn't expect someone with no background in audio at all to be reading this paper, and if that is the case one would expect the reader to look up the basics of the field -- space is just far too precious in a 9 page paper. I think it has a better balance now.
> >
> > I will update my score to 7, conditional on the remaining changes being made before camera ready -- I appreciate these can take longer than the rebuttal period.

---

### Official Review · Reviewer_4H7F · 2021-09-19
**Review of WaveFake paper**

**Rating:** 6
**Confidence:** 4
**Clarity:** The paper is clear and well-written.

**Strengths:**

The proposed dataset is a contribution to an important problem and might potentially foster the research on audio fake detection. The dataset is good for the standardization of deep fake research. The paper is generally clear and well-written.

**Weaknesses:**

1. It looks like creating such a dataset is relatively easy. The authors just took a couple of existing datasets and apply existing vocoders to them. Releasing this dataset might be valuable for the community, but every user might recreate it without too much effort. Maybe more than releasing the dataset, it could make sense to release a modular code that users can use to plug their model/datasets and create the distorted spectrograms (maybe this is available in https://github.com/RUB-SysSec/WaveFake, but I cannot see any repository at the given link).

2. The other limitation is that the datasets focus on 5 types of models only, but in the future, there might be many other models coming out and this dataset approach doesn't seem scalable. The popular HiFi-GAN, for instance, is missing.

3. It looks like the proposed GMM baseline is a bit outdated (proposed at Interspeech 2015). This is used as a baseline in the ASVspoof challenges, but it could be great to see more advanced baselines (e.g. systems based on neural nets).

Minor:

4. Signal processing recap: Is it really necessary for this paper? The reported information can be easily found somewhere and cannot be claimed as a contribution of this paper.


5. Fig.2 => add the same markers in the MelGAN figure to make the comparison easier.


**Additional Feedback:**

See above

**Correctness:**

The evaluation seems correct to me. My only concern is the adoption of a simple GMM baseline only.

**Documentation:**

I don't see issue on that parts.

**Ethics:**

No issues. I agree with the statement written by the authors.

**Relation To Prior Work:**

Prior work is described well.

**Summary And Contributions:**

The main contribution of this paper is the *WaveFake* datasets. It is created from LJSPEECH and JSUT using 5 different vocoders. The authors evaluated and analyzed the performance using a GMM-based system.

---

> ### Author Response · Authors · 2021-09-30
> **Reviewer 4H7F**
>
> We thank the reviewer for the detailed review and very engaging thoughts. We have made several changes in response to the review:
>
> > It looks like creating such a dataset is relatively easy. The authors just took a couple of existing datasets and apply existing vocoders to them. Releasing this dataset might be valuable for the community, but every user might recreate it without too much effort. Maybe more than releasing the dataset, it could make sense to release a modular code that users can use to plug their model/datasets and create the distorted spectrograms (maybe this is available in [https://github.com/RUB-SysSec/WaveFake](https://github.com/RUB-SysSec/WaveFake), but I cannot see any repository at the given link).
>
> As outlined above, we are sorry for the confusion. The code is attached to the openreview submission and will be moved to the public repository eventually.
> The code does contain scripts to reproduce all plots/statistics in the paper. The user has to simply supply a directory/file to analyze.
>
> > The other limitation is that the datasets focus on 5 types of models only, but in the future, there might be many other models coming out, and this dataset approach doesn't seem scalable. The popular HiFi-GAN, for instance, is missing.
>
> We have included samples from HiFi-GAN in our data set. Please note that we are waiting to update the Zenodo data set until the the review process is completed.
>
> > It looks like the proposed GMM baseline is a bit outdated (proposed at Interspeech 2015). This is used as a baseline in the ASVspoof challenges, but it could be great to see more advanced baselines (e.g., systems based on neural nets).
>
> We have included a neural network-based baseline model. The results are mixed. While the networks outperform LFCC-GMMs on the single and out-of-distribution task, the networks suffer a heavy penalty when evaluated in the phone simulation.
>
> > Signal processing recap: Is it really necessary for this paper? The reported information can be easily found somewhere and cannot be claimed as a contribution of this paper.
>
> We have removed the corresponding claim and reduced the related section.
>
> > Fig.2 => add the same markers in the MelGAN figure to make the comparison easier.
>
> We added the markers to the figure.

---

> > ### Comment · Reviewer_4H7F · 2021-09-30
> > **thanks to the authors!**
> >
> > I took a look at the changes and it looks like now the paper is in a better shape. Most of my comments have been addressed (e.g, HiFi GAN and adding a neural baseline). I increased my score.

---

### Official Review · Reviewer_XH3z · 2021-09-20
**A useful dataset for studying audio deepfake detection**

**Rating:** 7
**Confidence:** 3
**Correctness:** The claims seem to be correct.

**Strengths:**

Deepfakes, images and speech, can cause significant harm and are considered one of the main existing threats resulting from advances in AI. It is therefore important to study techniques to detect them. There has been extensive work, including the creation of datasets, in the area of fake images, yet not as much for audio. If the work presented here indeed consists of the first such dataset then I think that it is an important contribution (disclaimer: speech is not my field of research so there could be developments that I am not aware of). The authors should also be commended for the clear introduction, detailed background on this topic, and interesting experimental work.

**Weaknesses:**

I think that the main weakness, which the authors acknowledge, is the limited diversity of the training set - only two (female) speakers one speaking in English and the other in Japanese. This obviously calls for expanding this work to have significantly more diversity in voices, languages and recording conditions.

**Additional Feedback:**

A couple of typos:
-	Line 273: believe -> belief
-	Line 314: know -> now


**Clarity:**

The paper is very clear including an excellent background for readers who are not familiar with the topic.

**Documentation:**

Seems to be adequate.

**Ethics:**

The authors touch upon the ethical question of whether their work could contribute to an arms race between deepfake technology and techniques to detect them. It's a valid concern but I agree with the authors that in order to address the problem of deepfakes, researchers must have datasets to work with.

**Relation To Prior Work:**

Prior work seems to be well covered.

**Summary And Contributions:**

The paper describes a new dataset for studying audio (speech) deepfake detection. Specifically, they trained several deep generative models on existing speech datasets and sampled outputs from these models. The challenge is to distinguish the generated speech segments from the original ones. They also trained baseline models and ran some experiments to understand the types of approaches and statistics that look promising in detecting audio deepfake both against samples from the training set as well as out-of-distribution samples such as those coming from a different speaker, a different language, or including novel phrases.

---

> ### Author Response · Authors · 2021-09-30
> **Reviewer XH3z**
>
> We thank the reviewer for his time and appreciation of our work.
> We agree that our contribution is limited, and we do not claim to solve the problems of audio Deepfake datasets once and for all but instead wanted to provide a first benchmark data set in this underrepresented field.
> Since there is very little prior work on this subject.
> Thus, we opted for a broader approach.
> Besides the data set, we tried to give the (more vision-focused) community the resources to investigate audio media.
> However, a broader approach sadly implies that the individual parts have to be somewhat limited.
> We think more diverse data sets should be considered and believe this is an interesting direction for future work.

---

### Author Response · Authors · 2021-09-30
**High-Level Summary**

We want to thank our reviewers for their encouraging feedback and appreciation of our work. Based on the feedback received, we have made several changes to the manuscript. Here we provide a high-level summary and provide more detailed feedback within the specific comments. We uploaded an updated version of the manuscript with the changes highlighted in orange:

- We, unfortunately, forgot to remove the public GitHub link where the code will eventually get published. Please note that it is our lab's policy to not upload code while the manuscript is still under review. Our first submission included a corresponding disclaimer. However, we removed it for the first round's revision. The code is attached to the submission. Our pretrained models can be found [here](https://ruhr-uni-bochum.sciebo.de/s/Aqw3PKXMZMbWBPE) and will eventually be moved to Zenodo.

- As suggested by reviewer 4H7F we have extended our data set with samples from HiFi-GAN. We also updated the corresponding datasheet to reflect better the question-based format used by Datasheets for Datasets.

- We have extended our baseline experiments with a neural baseline as suggested by reviewer 4H7F. We used the RawNet2 model, which is also used by the current iteration of the ASVspoof workshop (2021). Please note that this baseline was announced after our initial submission to the best of our knowledge. In previous challenges, only the GMM baselines were used.

- Due to the inclusion of the HiFi-GAN samples, we had to redo our earlier experiments. Unfortunately, this meant that the RawNet2 out-of-distribution experiments only finished for 3/7 networks. However, the results look consistent, and we reported them to the paper. We will add the final results to the camera-ready version.

- Multiple reviewers have pointed out that the signal processing recap is not a contribution in its own right. We agree and have removed this part from our claims. However, in the first round of reviews, this section was especially praised for being helpful to the greater ML/NeurIPS community since they are probably less familiar with audio processing. Thus, we have reduced the section only to describe the different feature representations used throughout the paper.

- We utilized the gained space to extend other underdeveloped sections pointed out in the reviews. Note that we do not specifically highlight grammar or typo fixes.

---

### Decision · Program_Chairs · 2021-10-09

**Decision:**

Accept

**Comment:**

The data provided in the paper is of relevance for the NeurIPS data track. Based on reviewers opinions, and in particular after discussion with the authors, the paper achieves the minimum score required for publication at NeurIPS data track.